# DEEP VARIATIONAL SEMI-SUPERVISED NOVELTY DETECTION

## ABSTRACT

In anomaly detection (AD), one seeks to identify whether a test sample is abnormal, given a data set of normal samples. A recent and promising approach to AD relies on deep generative models, such as variational autoencoders (VAEs), for unsupervised learning of the normal data distribution. In semi-supervised AD (SSAD), the data also includes a small sample of labeled anomalies. In this work, we propose two variational methods for training VAEs for SSAD. The intuitive idea in both methods is to train the encoder to 'separate' between latent vectors for normal and outlier data. We show that this idea can be derived from principled probabilistic formulations of the problem, and propose simple and effective algorithms. Our methods can be applied to various data types, as we demonstrate on SSAD datasets ranging from natural images to astronomy and medicine, and can be combined with any VAE model architecture. When comparing to state-of-the-art SSAD methods that are not specific to particular data types, we obtain marked improvement in outlier detection.

## 1 INTRODUCTION

Anomaly detection (AD) – the task of identifying samples that are abnormal with respect to some normal data – has important applications in domains ranging from health-care, to security, and robotics (Pimentel et al., 2014). In its common formulation, training data is provided only for normal samples, while at test time, anomalous samples need to be detected. In the probabilistic AD approach, a model of the normal data distribution is learned, and the likelihood of a test sample under this model is thresholded for classification as normal or not. Recently, deep generative models such as variational autoencoders (VAEs, Kingma & Welling 2013) and generative adversarial networks (Goodfellow et al., 2014) have shown promise for learning data distributions in AD (An & Cho, 2015; Suh et al., 2016; Schlegl et al., 2017; Wang et al., 2017).

Here, we consider the setting of *semi-supervised* AD (SSAD), where in addition to the normal samples, a small sample of labeled anomalies is provided (Görnitz et al., 2013). Most importantly, *this set is too small to represent the range of possible anomalies*, making classification methods (either supervised or semi-supervised) unsuitable. Instead, most approaches are based on 'fixing' an unsupervised AD method to correctly classify the labeled anomalies, while still maintaining AD capabilities for unseen outliers (e.g., Görnitz et al., 2013; Muñoz-Marí et al., 2010; Ruff et al., 2019).

In this work, we present a variational approach for learning data distributions in the SSAD problem setting. We base our method on the VAE, and modify the training objective to account for the labeled outlier data. We propose two formulations for this problem. The first maximizes the log-likelihood of normal samples, while minimizing the log-likelihood of outliers, and we effectively optimize this objective by combining the standard evidence lower bound (ELBO) with the $\chi$ upper bound (CUBO, Dieng et al. 2017). The second method is based on separating the VAE prior between normal and outlier samples. Effectively, both methods have a similar intuitive interpretation: they modify the VAE encoder to push outlier samples away from the prior distribution (see Figure 1). Importantly, our method does not place any restriction on the VAE architecture, and can be used to modify any VAE to account for outliers. As such, it can be used for general types of data.

We evaluate our methods in the comprehensive SSAD test-suite of Ruff et al. (2019), which includes both image data and low-dimensional data sets from astronomy, medicine, and other domains, and report a marked improvement in performance compared to both shallow and deep methods. In

addition, we demonstrate the flexibility of our method by modifying a conditional VAE used for generating sampling distributions for robotic motion planning (Ichter et al., 2018) to not generate way points that collide with obstacles.

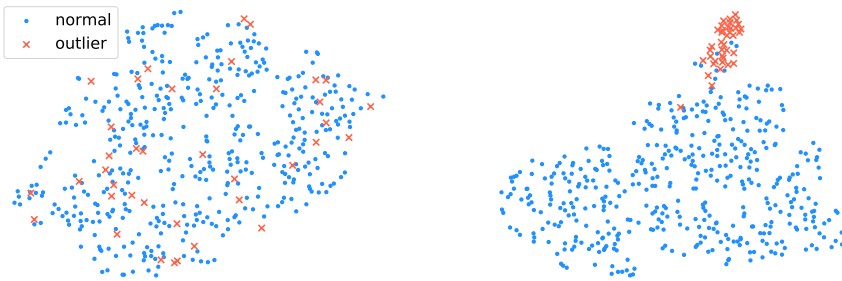

Figure 1: t-SNE of the latent space of a conventional VAE (left), and our proposed dual-prior VAE (right), trained on the Thyroid dataset, and evaluated on an unseen test sample. Note that latent vectors for outliers (red crosses) in DP-VAE are pushed away from the normal samples.

## 2  RELATED WORK

Anomaly detection, a.k.a. outlier detection or novelty detection, is an active field of research. Shallow methods such as one-class SVM (OC-SVM, Schölkopf et al. 2001) or support vector data description (SVDD, Tax & Duin 2004) have shown great success, but rely on hand-crafted features. Recently, there is growing interest in deep learning methods for AD (Chalapathy & Chawla, 2019).

Most studies on semi-supervised AD also require hand designed features. Muñoz-Marí et al. (2010) proposed $S^2$OC-SVM, a modification of OC-SVM that introduces labeled and unlabeled samples, Görnitz et al. (2013) proposed an approach based on SVDD, and Blanchard et al. (2010) base their approach on statistical testing. Deep SSAD has been studied recently in specific contexts such as videos (Kiran et al., 2018), network intrusion detection (Min et al., 2018), or specific neural network architectures (Ergen et al., 2017). The most relevant prior work to our study is the recently proposed deep semi-supervised anomaly detection (Deep SAD) approach of Ruff et al. (2019). Deep SAD is a general method based on deep SVDD (Ruff et al., 2018), which learns a neural-network mapping of the input that minimizes the volume of data around a predetermined point. While it has been shown to work on a variety of domains, Deep SAD must place restrictions on the network architecture, such as no bias terms (in all layers) and no bounded activation functions, to prevent degeneration of the minimization problem to a trivial solution. The methods we propose here do not place any restriction on the network architecture, and can be combined with any VAE model. In addition, we show improved performance in almost all domains compared to Deep SAD's state-of-the-art results.

While our focus here is SSAD, for completeness we review recent work on deep AD. Erfani et al. (2016); Andrews et al. (2016); Cao et al. (2016); Chen et al. (2017) follow a hybrid approach where deep unsupervised learning is used to learn features, which are then used within a shallow AD method. Specifically for image domains, Golan & El-Yaniv (2018) learn features using a self-supervised paradigm – by applying geometric transformations to the image and learning to classify which transformation was applied. Lee et al. (2018) learn a distance metric in feature space, for networks pre-trained on image classification. Both of these methods display outstanding AD performance, but are limited to image domains, while our approach does not require particular data or pre-trained features. Several studies explored using deep generative models such as GANs and VAEs for AD (e.g., An & Cho, 2015; Suh et al., 2016; Schlegl et al., 2017; Wang et al., 2017), and more recently, deep energy-based models (Zhai et al., 2016; Song & Ou, 2018) have reported outstanding results. Our work extends the VAE approach to the SSAD setting. Our results show that even a 1% fraction of labelled anomalies can improve over the state-of-the-art AD scores of deep energy based models, demonstrating the importance of the SSAD setting.

## 3 BACKGROUND

We consider deep unsupervised learning under the standard variational inference setting (Kingma & Welling, 2013). Given some data $x$, one aims to fit the parameters $\theta$ of a latent variable model $p_\theta(x) = \mathbb{E}_{p(z)}[p_\theta(x|z)]$, where the prior $p(z)$ is known. For general models, the typical maximum-likelihood objective $\max_\theta \log p_\theta(x)$ is intractable due to the marginalization over $z$, and can be approximated using the following variational inference methods.

### 3.1 EVIDENCE LOWER BOUND (ELBO)

The evidence lower bound (ELBO) states that for some approximate posterior distribution $q(z|x)$:

$$\log p_\theta(x) \geq \mathbb{E}_{q(z|x)}\left[\log p_\theta(x|z)\right] - D_{\mathrm{KL}}(q(z|x)\|p(z)),$$

where the Kullback-Leibler divergence is $D_{\mathrm{KL}}(q(z|x)\|p(z)) = \mathbb{E}_{q(z)}\left[\log \frac{q(z|x)}{p(z)}\right]$. In the variational autoencoder (VAE, Kingma & Welling 2013), the approximate posterior is represented as $q_\phi(z|x) = \mathcal{N}(\mu_\phi(x), \Sigma_\phi(x))$ for some neural network with parameters $\phi$, the prior is $p(z) = \mathcal{N}(\mu_0, \Sigma_0)$, and the ELBO can be maximized using the *reparameterization trick*. Since the resulting model resembles an autoencoder, the approximate posterior $q_\phi(z|x)$ is also known as the *encoder*, while $p_\theta(x|z)$ is termed the *decoder*.

### 3.2 $\chi$ UPPER BOUND (CUBO)

Recently, Dieng et al. (2017) derived variational upper bounds to the data log-likelihood. The $\chi^n$-divergence is given by $D_{\chi^n}(p\|q) = \mathbb{E}_{q(z;\theta)}\left[\left(\frac{p(z|X)}{q(z;\theta)}\right)^n - 1\right]$. For $n > 1$, Dieng et al. (2017) propose the following $\chi$ upper bound (CUBO):

$$\log p_\theta(x) \leq \frac{1}{n}\log \mathbf{E}_{q(z|x)}\left[\left(\frac{p_\theta(x|z)p(z)}{q(z|x)}\right)^n\right] \doteq CUBO_n(q(z|x)).$$

For a family of approximate posteriors $q_\phi(z|x)$, one can minimize the $CUBO$ using Monte Carlo (MC) estimation. However, MC gives a lower bound to CUBO and its gradients are biased. As an alternative, Dieng et al. (2017) proposed the following optimization objective: $\mathcal{L} = \exp\{n \cdot CUBO_n(q_\phi(z|x))\}$. By monotonicity of $\exp$, this objective reaches the same optima as $CUBO_n(q_\phi(z|x))$. Moreover, this produces an unbiased estimate, and the number of samples only affects the variance of the gradients.

## 4 DEEP VARIATIONAL SEMI-SUPERVISED ANOMALY DETECTION

In the anomaly detection problem, the goal is to detect whether a sample $x$ was generated from some normal distribution $p_{normal}(x)$, or not – making it an anomaly. In semi-supervised anomaly detection (SSAD), we are given $N_{normal}$ samples from $p_{normal}(X)$, which we denote $X_{normal}$. In addition, we are given $N_{outlier}$ examples of anomalous data, denoted by $X_{outlier}$, and we assume that $N_{outlier} \ll N_{normal}$. In particular, $X_{outlier}$ does not cover the range of possible anomalies, and thus classification methods (neither supervised nor semi-supervised) are not applicable.

Our approach for SSAD is to approximate $p_{normal}(x)$ using a deep latent variable model $p_\theta(x)$, and to decide whether a sample is anomalous or not based on thresholding its predicted likelihood. In the following, we propose two variational methods for learning $p_\theta(x)$. The first method, which we term *max-min likelihood* VAE (MML-VAE), maximizes the likelihood of normal samples while minimizing the likelihood of outliers. The second method, which we term *dual-prior* VAE (DP-VAE), assumes different priors for normal and outlier samples.

### 4.1 MAX-MIN LIKELIHOOD VAE

In this approach, we seek to find model parameters based on the following objective:

$$\max_\theta \quad \log p_\theta(X_{normal}) - \gamma \log p_\theta(X_{outlier}), \tag{1}$$

where $\gamma \geq 0$ is a weighting term. Note that, in the absence of outlier data or when $\gamma = 0$, Equation 1 is just maximum likelihood estimation. For $\gamma > 0$, however, we take into account the knowledge that outliers should not be assigned high probability.

We model the data distribution using a latent variable model $p_\theta(x) = \mathbb{E}_{p(z)}[p_\theta(x|z)]$, where the prior $p(z)$ is known. To optimize the objective in Equation 1 effectively, we propose the following variational lower bound:

$$\log p_\theta(X_{normal}) - \gamma \log p_\theta(X_{outlier}) \geq ELBO_{Q_1}(X_{normal}) - \gamma CUBO_{Q_2}(X_{outlier}), \quad (2)$$

where $Q_1(z|x), Q_2(z|x)$ are the variational auxiliary distributions. In principle, the objective in Equation 2 can be optimized using the methods of Kingma & Welling (2013) and Dieng et al. (2017), which would effectively require training two encoders: $Q_1$ and $Q_2$, and one decoder $p_\theta(x|z)$,[1] separately on the two datasets. However, it is well-known that training deep generative models requires abundant data. Thus, there is little hope for learning an informative $Q_2$ using the small dataset $X_{outlier}$. To account for this, our main idea is to use the same variational distribution $Q(z|x)$ for both loss terms, which effectively relaxes the lower bound as follows:

$$\log p_\theta(X_{normal}) - \gamma \log p_\theta(X_{outlier}) \geq ELBO_Q(X_{normal}) - \gamma CUBO_Q(X_{outlier}). \quad (3)$$

In other words, we use the **same encoder** for both normal and anomalous data. Finally, the loss function of the MML-VAE is: $\mathcal{L} = \gamma CUBO_Q(X_{outlier}) - ELBO_Q(X_{normal})$.

To gain intuition about the effect of the CUBO term in the loss function, it is instructive to assume that $p_\theta(x|z)$ is fixed. In this case, maximizing the lower bound only affects the variational distribution $q(z|x)$. Note that the ELBO term seeks to minimize the KL distance between $q(z|X_{normal})$ and $p(z|X_{normal})$, which 'pushes' $q(z|X_{normal})$ toward high-likelihood regions of $p(z)$, while the CUBO term is proportional to $\frac{1}{n} \log \mathbb{E}_{q(z|x)} \left[ \left( \frac{p(z|X_{outlier})}{q(z|X_{outlier})} \right)^n \right]$, and thus seeks to maximize the $\chi_n$ distance between $q(z|X_{outlier})$ and $p(z|X_{outlier})$. Now, assume that for some outlier, $p(z|X_{outlier})$ falls within a high-likelihood region of $p(z)$. In this case, the CUBO term will 'push' $q(z|X_{outlier})$ away from $p(z)$. Thus, intuitively, the CUBO term seeks to **separate** the latent distributions for normal samples, which will concentrate on high-likelihood regions of $p(z)$, and outlier samples.

For the common choice of Gaussian prior and approximate posterior distributions (Kingma & Welling, 2013), the CUBO component of the loss function is given by:

$$\mathcal{L}_{CUBO} = \exp\{-2 \cdot \mathcal{L}_R + \log|\Sigma_q| + \mu_q^T \Sigma_q^{-1} \mu_q +$$
$$\log \mathbb{E}_q \left[ \exp\{-z^T z + z^T \Sigma_q^{-1} z - 2z^T \Sigma_q^{-1} \mu_q\} \right] \}.$$

Where $\mathcal{L}_R = -\log p_\theta(x|z)$ is the reconstruction error. The expectation is approximated with MC. In our experiments, we found that computing $\exp(\log \mathbb{E}[\exp(\cdot)])$ is numerically more stable, as we can employ the log-sum-exp trick.

In practice, we found that updating only the encoder according to the CUBO loss (i.e., ignoring the $\mathcal{L}_R$ term) results in better performance and stabler training. The reason for this is that CUBO seeks to maximize the reconstruction error, which does not produce informative updates for the decoder. Since this decoder is shared with the ELBO term, the CUBO update can decrease performance for normal samples as well. Complete algorithm details and derivation of the CUBO objective are provided Appendix A.4.

### 4.2 DUAL PRIOR VAE

The second method we propose is a simple yet effective modification of the VAE for SSAD, which we term dual-prior VAE (DP-VAE). Here, we assume that both normal and outlier data are generated from a single latent variable model $p_\theta(x) = \mathbb{E}_{p(z,y)}[p_\theta(x|z,y)]$, where $y$ is an additional Bernoulli variable that specifies whether the sample is normal or not. Additionally, we assume the following conditional prior for $z$:[2]

$$p(z|y) = \begin{cases} \mathcal{N}(\mu_{normal}, \Sigma_{normal}), & y = 1 \\ \mathcal{N}(\mu_{outlier}, \Sigma_{outlier}), & y = 0 \end{cases}.$$

---

[1] Based on the ELBO and CUBO definitions, the decoder $p_\theta(x|z)$ is the same for both terms.

[2] This method can easily be extended to non-Gaussian priors, so long as the two priors are different.

Since our data is labeled, $y$ is known for every $x$ and does not require inference. Deriving the ELBO for normal and outlier samples in this case gives:

$$ELBO_{normal}(x) = -\mathcal{L}_R(x) - D_{\text{KL}}[Q(z|x)\|p(z|y=1)],$$

$$ELBO_{outlier}(x) = -\mathcal{L}_R(x) - D_{\text{KL}}[Q(z|x)\|p(z|y=0)].$$

Here, similarly to the MML-VAE above, we assume *the same encoder* $Q(z|X)$ for normal and outlier samples. The loss function we minimize is

$$\mathcal{L} = -\left[ELBO_{normal}(X_{normal}) + ELBO_{outlier}(X_{outlier})\right].$$

In our implementation, we set $\Sigma_{normal} = \Sigma_{outlier} = I$, $\mu_{normal} = 0$, and $\mu_{outlier} = \alpha I$ where $\alpha \neq 0$. In this case, this loss function has the intuitive interpretation of training an encoder that separates between latent variables for normal and outlier data.

As $\mathcal{L}$ is simply a combination of VAE loss functions, optimization using the reparametrization trick is straightforward.[3] Similarly to the MML-VAE method, we freeze the decoder when minimizing $ELBO_{outlier}$, as we do not expect for an informative reconstruction of the outlier distribution from the limited data $X_{outlier}$.

### 4.3 ANOMALY DETECTION WITH MML-VAE AND DP-VAE

We next describe an SSAD algorithm based on MML-VAE and DP-VAE. After training an MML-VAE, the $ELBO$ provides an approximation of the normal data log-likelihood. Thus, given a novel sample $x$, we can score its (log) likelihood of belonging to normal data as:

$$score(x) = ELBO(x).$$

After training a DP-VAE, $ELBO_{normal}$ provides a similar approximation, and in this case we set $score(x) = ELBO_{normal}(x)$. Finally, for classifying the sample, we threshold its score.

At this point, the reader may wonder why we pick the ELBO and not the CUBO for scoring a sample. The reason is that most of our data is used for training the ELBO, while the CUBO term (and $ELBO_{outlier}$ in DP-VAE) is trained with a much smaller outlier data, which is not informative enough to accurate approximate the normal data likelihood.

### 4.4 VAE ARCHITECTURES AND ENSEMBLES

Both MML-VAE and DP-VAE simply add loss terms to the objective function of the conventional VAE. Thus, they can be applied to any type of VAE without any restriction on the architecture. In addition, they can be used with ensembles of VAEs, a popular technique to robustify VAE training (Chen et al., 2017; Mirsky et al., 2018). We train $K$ different VAEs (either MML-VAE or DP-VAE) with random initial weights, and set the score to be the average over the ELBOs in the ensemble. We found that ensembles significantly improve the SSAD performance of our methods, as demonstrated in our experiments.

## 5 EXPERIMENTS

In our experiments, we follow the evaluation methods and datasets proposed in the extensive work of Ruff et al. (2019). These evaluations include strong baselines of state-of-the-art shallow, deep, and hybrid algorithms for AD and SSAD. Hybrid algorithms are defined as shallow SSAD methods that are trained on features extracted from a deep autoencoder trained on the raw data. A brief overview of the methods is provided in Appendix A.2, and we refer the reader to Ruff et al. (2019) for a more detailed description. Performance is evaluated by the area under the curve of the receiver operating characteristic curve (AUROC), a commonly used criterion for AD. There are two types of datasets: (1) high-dimensional datasets which were modified to be semi-supervised and (2) classic anomaly detection benchmark datasets. The first type includes MNIST, Fashion-MNIST and CIFAR-10, and

---

[3]We also experimented with a hybrid model, trained by combining the loss functions for MML-VAE and DP-VAE, and can be seen as a DP-VAE with the additional loss of minimizing the likelihood of outlier samples in $ELBO_{normal}(x)$. This model obtained similar performance, as we report in the appendix.

the second includes datasets from various fields such as astronomy and medicine. For strengthening the baselines, Ruff et al. (2019) grant the shallow and hybrid methods an unfair advantage of selecting their hyper-parameters to maximize AUROC on a subset (10%) of the test set. Here we also follow this approach. When comparing with the state-of-the-art Deep SAD method, we used the same network architecture but include bias terms in all layers. For the MNIST dataset, we found that this architecture did not work well, and instead used a standard convolutional neural network (CNN) architecture (see appendix for full details). Our implementation is done in PyTorch (Paszke et al., 2017) and we run our experiments on a machine with an Nvidia RTX 2080 GPU. For all of the experiments, our methods use an ensemble of size $K = 5$.

## 5.1 IMAGE DATASETS

**MNIST, Fashion-MNIST and CIFAR-10**   datasets all have ten classes. Similarly to Ruff et al. (2019), we derive ten AD setups on each dataset. In each setup, one of the ten classes is set to be the normal class, and the remaining nine classes represent anomalies. During training, we set $X_{normal}$ as the normal class samples, and $X_{outlier}$ as a small fraction of the data from **only one** of the anomaly classes. At test time, we evaluate on anomalies from **all** anomaly classes. For pre-processing, pixels are scaled to $[0, 1]$. Unlike Ruff et al. (2019) which did not report on using a validation set to tune the hyper-parameters, we take a validation set (20%) out of the training data to tune hyper-parameters.

The experiment we perform, similarly to Ruff et al. (2019), is evaluating the model's detection ability as a function of the ratio of anomalies presented to it during training. We set the ratio of labeled training data to be $\gamma_l = N_{outlier}/(N_{normal} + N_{outlier})$, and we evaluate different values of $\gamma_l$ in each scenario. In total, there are 90 experiments per each value of $\gamma_l$. Note that for $\gamma_l = 0$, no labeled anomalies are presented, and we revert to standard *unsupervised* AD, which in our approach amounts to training a standard VAE ensemble.

Our results are presented in Table 1. The complete table with all of the competing methods can be found in the appendix A.1.1. Note that even a small fraction of labeled outliers ($\gamma_l = 0.01$) leads to significant improvement compared to the standard unsupervised VAE ($\gamma_l = 0$) and compared to the best-performing unsupervised methods on MNIST and CIFAR-10. Also, our methods outperform other SSAD baselines in most domains.

| Data | $\gamma_l$ | OC-SVM Raw | OC-SVM Hybrid | Inclusive NRF | SSAD Raw | SSAD Hybrid | Deep SAD | Supervised Classifier | MML VAE | DP VAE |
|---|---|---|---|---|---|---|---|---|---|---|
| MNIST | .00 | 96.0±2.9 | **96.3±2.5** | 95.2±3.0 | 96.0±2.9 | 96.3±2.5 | 92.8±4.9 | | 94.2±3.0 | 94.2±3.0 |
| | .01 | | | | 96.6±2.4 | 96.8±2.3 | 96.4±2.7 | 92.8±5.5 | **97.3±2.1** | 97.0±2.3 |
| | .05 | | | | 93.3±3.6 | 97.4±2.0 | 96.7±2.4 | 94.5±4.6 | **97.8±1.6** | 97.5±2.0 |
| | .10 | | | | 90.7±4.4 | 97.6±1.7 | 96.9±2.3 | 95.0±4.7 | **97.8±1.6** | 97.6±2.1 |
| | .20 | | | | 87.2±5.6 | 97.8±1.5 | 96.9±2.4 | 95.6±4.4 | **97.9±1.6** | **97.9±1.8** |
| F-MNIST | .00 | **92.8±4.7** | 91.2±4.7 | | **92.8±4.7** | 91.2±4.7 | 89.2±6.2 | | 90.8±4.6 | 90.8±4.6 |
| | .01 | | | | **92.1±5.0** | 89.4±6.0 | 90.0±6.4 | 74.4±13.6 | 91.2±6.6 | 90.9±6.7 |
| | .05 | | | | 88.3±6.2 | 90.5±5.9 | 90.5±6.5 | 76.8±13.2 | 91.6±6.3 | **92.2±4.6** |
| | .10 | | | | 85.5±7.1 | 91.0±5.6 | 91.3±6.0 | 79.0±12.3 | **91.7±6.4** | **91.7±6.0** |
| | .20 | | | | 82.0±8.0 | 89.7±6.6 | 91.0±5.5 | 81.4±12.0 | 91.9±6.0 | **92.1±5.7** |
| CIFAR-10 | .00 | 62.0±10.6 | 63.8±9.0 | **70.0±4.9** | 62.0±10.6 | 63.8±9.0 | 60.9±9.4 | | 52.7±10.7 | 52.7±10.7 |
| | .01 | | | | 73.0±8.0 | 70.5±8.3 | 72.6±7.4 | 55.6±5.0 | 73.7±7.3 | **74.5±8.4** |
| | .05 | | | | 71.5±8.1 | 73.3±8.4 | 77.9±7.2 | 63.5±8.0 | **79.3±7.2** | 79.1±8.0 |
| | .10 | | | | 70.1±8.1 | 74.0±8.1 | 79.8±7.1 | 67.7±9.6 | 80.8±7.7 | **81.1±8.1** |
| | .20 | | | | 67.4±8.8 | 74.5±8.0 | 81.9±7.0 | 80.5±5.9 | 82.6±7.2 | **82.8±7.3** |

Table 1: Results for image datasets. We report the average and standard deviation of AUROC over 90 experiments, for various ratios of labeled anomalies in the data $\gamma_l$.

**CatsVsDogs Dataset**   In addition to the test domains of Ruff et al. (2019), we also evaluate on the CatsVsDogs dataset, which is notoriously difficult for anomaly detection (Golan & El-Yaniv, 2018). This dataset contains 25,000 images of cats and dogs in various positions, 12,500 in each class. Following Golan & El-Yaniv (2018), We split this dataset into a training set containing 10,000 images, and a test set of 2,500 images in each class. We also rescale each image to size 64x64. We follow a similar experimental procedure as described above, and average results over the two classes. We chose a VAE architecture similar to the autoencoder architecture in Golan & El-Yaniv (2018), and for the *Deep SAD* baseline, we modified the architecture to not use bias terms and

bounded activations. We report our results in Table 2, along with the numerical scores for baselines taken from Golan & El-Yaniv (2018). Note that without labeled anomalies, our method is not informative, predicting roughly at chance level, and this aligns with the baseline results reported by Golan & El-Yaniv (2018). However, even just 1% labeled outliers is enough to significantly improve predictions and produce informative results. *This demonstrates the potential of the SSAD approach.* However, in this domain, using the geometric transformations of Golan & El-Yaniv (2018) allow for significantly better performance even without labeled outliers. While this approach is domain specific, incorporating similar self-supervised learning ideas into probabilistic AD methods is an interesting direction for future research.

| Data | $\gamma_l$ | OC-SVM Raw | OC-SVM Hybrid | DAGMM | DSEBM | ADGAN | DADGT | Deep SAD | MML VAE | DP VAE |
|---|---|---|---|---|---|---|---|---|---|---|
| CatsVsDogs | .00 | 51.7 | 52.5 | 47.7 | 51.6 | 49.4 | 88.8 | 49.9 | 50.7 | 50.7 |
| | .01 | | | | | | | 54.1 | 59.4 | 64.0 |
| | .05 | | | | | | | 60.1 | 68.4 | 70.3 |
| | .10 | | | | | | | 64.4 | 71.5 | 75.3 |
| | .20 | | | | | | | 67.2 | 73.3 | 78.2 |

Table 2: Results of the experiment where we increase the ratio of labeled anomalies $\gamma_l$ in the training set, on the CatsVsDogs dataset.

## 5.2 CLASSIC ANOMALY DETECTION DATASETS

AD is a well-studied field of research, and as a result, there are many publicly available AD benchmarks (Shebuti, 2016) with well-established baselines. These datasets are lower-dimensional than the image datasets above, and by evaluating our method on them, we aim to demonstrate the flexibility of our method for general types of data. We follow Ruff et al. (2019), and consider random train-test split of 60:40 with stratified sampling to ensure correct representation of normal and anomaly classes. We evaluate over ten random seeds with 1% of anomalies, i.e., $\gamma_l = 0.01$. There are no specific anomaly classes, thus we treat all anomalies as one class. For pre-processing we preform standardization of the features to have zero mean and unit variance. As most of these datasets have a very small amount of labeled anomalous data, it is inefficient to take a validation set from the training set. To tune the hyper-parameters, we measure AUROC performance on the training data, and we also follow Ruff et al. (2019) and set a 150 training epochs limit. The complete hyper-parameters table can be found in Appendix A.3.2.

In Table 3 we present the results of the best-performing algorithms along with our methods. The complete table with all the competing methods can be found in Appendix A.1.2. Our methods outperform all of the other methods on most datasets, demonstrating the flexibility of our approach for general types of data.

| Dataset | OC-SVM Raw | SSAD Raw | Supervised Classifier | Deep SAD | VAE | MML VAE | DP VAE |
|---|---|---|---|---|---|---|---|
| arrhythmia | 84.5±3.9 | **86.7±4.0** | 39.2±9.5 | 75.9±8.7 | 85.6 ± 2.4 | 85.7 ± 2.3 | **86.7 ± 1.7** |
| cardio | 98.5±0.3 | 98.8±0.3 | 83.2±9.6 | 95.0±1.6 | 95.5 ± 0.8 | **99.2 ± 0.4** | 99.1 ± 0.4 |
| satellite | 95.1±0.2 | **96.2±0.3** | 87.2±2.1 | 91.5±1.1 | 77.7 ± 1.0 | 92.0 ± 1.5 | 89.2 ± 1.6 |
| satimage-2 | 99.4±0.8 | **99.9±0.1** | **99.9±0.1** | **99.9±0.1** | 99.6 ± 0.6 | 99.7 ± 0.3 | **99.9 ± 0.1** |
| shuttle | 99.4±0.9 | 99.6±0.5 | 95.1±8.0 | 98.4±0.9 | 98.1 ± 0.6 | 99.8 ± 0.1 | **99.9 ± 0.04** |
| thyroid | 98.3±0.9 | 97.9±1.9 | 97.8±2.6 | 98.6±0.9 | 86.9 ± 3.7 | **99.9 ± 0.04** | **99.9 ± 0.04** |

Table 3: Results on classic AD benchmark datasets in the setting with a ratio of labeled anomalies of $\gamma_l = 0.01$ in the training set. We report the avg. AUROC with st.dev. computed over 10 seeds.

## 5.3 ABLATIVE ANALYSIS

We perform an ablative analysis of DP-VAE method on the cardio, satellite and CIFAR-10 datasets (with 1% outliers). We evaluate our method with the following properties: (1) Frozen and unfrozen decoder, (2) separate and same encoder for normal data and outliers and (3) the effect of using ensembles of VAEs instead of one. Table 4 summarizes the analysis; the complete table can be found in Appendix A.2.1. It can be seen that using the same encoder is necessary as we expected, since

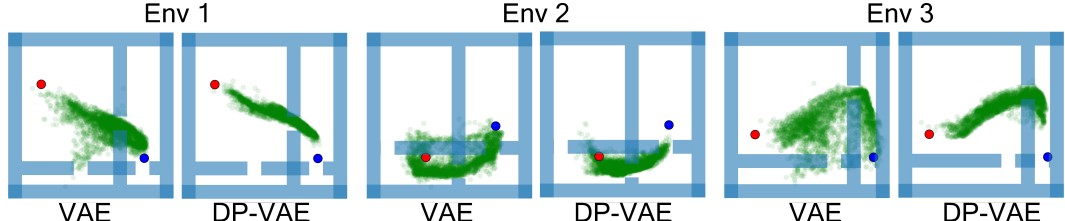

Figure 2: Motion planning with VAE. Following the work of Ichter et al. (2018), a conditional VAE is trained to generate robot configurations to be used within a sampling based motion planning algorithm. The VAE generates 6D robot configurations (robot position and velocity; here projected onto the 2D plane) given an image of obstacles, an initial state (blue) and a goal state (red). Using our DP-VAE method, we added to the VAE training negative samples on obstacle boundaries. We compare samples from the standard VAE (left) and DP-VAE (right) on three unseen obstacle maps. Note that our method results in much more informative samples for planning obstacle avoidance.

training a neural network requires sufficient amount of data. Moreover, when dealing with a small pool of outliers, the effect on the decoder is minimal. Hence, freezing the decoder contributes little to the improvement. Ensembles improve our methods result by approximately 2-4% on average, which makes sense as VAE training is a stochastic process. For a fair comparison with the state-of-the-art, we have also evaluated an ensemble of Deep SAD models. These results, which are detailed in Appendix A.1.1, show that ensembles have little effect on Deep SAD. This can be explained as follows. In Deep-SAD, confidence is measured according to distance to an arbitrary point $c$ in the vector space. Thus, the scores from different networks in the ensemble are not necessarily calibrated (they have different $c$ points). In our VAE approach, on the other hand, the score of each network is derived from the same principled variational bound, and therefore the scores are calibrated, giving rise to the benefit of the ensemble.

| Encoder | Decoder | Ensemble | Cardio AUROC | Satellite AUROC | CIFAR-10 AUROC |
|---------|---------|----------|--------------|-----------------|----------------|
| Separate | Unfreeze | 5 | 96.6±0.6 | 65.3±1.1 | 49.8±10.7 |
| Separate | Freeze | 5 | 96.6±0.6 | 66.9±1.2 | 50.0±10.9 |
| Same | Unfreeze | 5 | 98.8±0.06 | 88.9±1.7 | 73.9±8.9 |
| Same | Freeze | 5 | **99.1±0.4** | **91.7±1.5** | **74.5±8.4** |
| Same | Freeze | 1 | 97.8±1.1 | 87.6±1.7 | 72.3±9.0 |

Table 4: Ablative analysis of the Dual Prior method. AUROC is reported over average of 10 seeds for the *satellite* and *cardio*. For *CIFAR-10*, results are reported for the experiment with 1% of anomalies, averaged over 90 experiments.

## 5.4 SAMPLE-BASED MOTION PLANNING APPLICATION

While our focus is on SSAD, our methods can be used to enhance any VAE generative model when negative samples are available. We demonstrate this idea in a motion planning domain (Latombe, 2012). Sampling-based planning algorithms search for a path for a robot between obstacles on a graph of points sampled from the feasible configurations of the system. In order to accelerate the planning process, Ichter et al. (2018) suggest to learn non-uniform sampling strategies that concentrate in regions where an optimal solution might lie, given a training set of planning problems to learn from. Ichter et al. (2018) proposed a conditional VAE that samples configurations conditioned on an image of the obstacles in the domain, and is trained using samples of feasible paths on a set of training domains. Here, we propose to enhance the quality of the samples by including a few outliers (5% of the normal data) during training, which we choose to be points on the obstacle boundaries – as such points clearly do not belong to the motion plan. We used the publicly available code base of Ichter et al. (2018), and modified only the CVAE training using our DP-VAE method. Exemplary generated samples for different obstacle configurations (unseen during training) are shown in Figure 2. It can be seen that adding the outliers led to a VAE that is much more focused on the feasible parts of the space, and thus generates significantly less useless points in collision with obstacles.

## 6  CONCLUSION

We proposed two VAE modifications that account for negative data examples, and used them for semi-supervised anomaly detection. We showed that these methods can be derived from natural probabilistic formulations of the problem, and that the resulting algorithms are general and effective – they outperform the state-of-the-art on diverse datasets. We further demonstrated that even a small fraction of outlier data can significantly improve anomaly detection on various datasets, and that our methods can be combined with VAE applications such as in motion planning.

We see great potential in the probabilistic approach to AD using deep generative models: it has a principled probabilistic interpretation, it is agnostic to the particular type of data, and it can be implemented using expressive generative models. For specific data such as images, however, discriminative approaches that exploit domain specific methods such as geometric transformations are currently the best performers. Developing similar self-supervised methods for generative approaches is an exciting direction for future research. Another promising direction would be incorporating SSAD within energy-based models.

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

# A    APPENDIX

## A.1    COMPLETE RESULTS

### A.1.1    MNIST, FASHION-MNIST, CIFAR-10

Table 5 includes the complete results. The results for an ensemble of Deep SAD models are in parenthesis (CIFAR-10). The ensemble method is implemented the same way as ours: we train $K = 5$ separate models (i.e., each model has its own $c$), and the score (which is the distance from $c$) is the average of scores from all models in the ensemble.

### A.1.2    CLASSIC ANOMALY DETECTION

Table 6 includes the complete results. **MML-DP VAE** combines both suggested objectives when training the VAE, i.e. MML and DP, and as reflected in the results, its performance is on par with them.

## A.2    COMPETING METHODS

We compare our methods to reported performance of both deep and shallow learning approaches, as detailed by Ruff et al. (2019) and Golan & El-Yaniv (2018). For completeness, we give a brief overview of the methods.

**OC-SVM/SVDD**    The one-class support vector machine (OC-SVM) is a kernel based method for novelty detection (Schölkopf et al., 2001). It is typically employed with an RBF kernel, and learns a collection of closed sets in the input space, containing most of the training samples. SVDD (Tax & Duin, 2004) is equivalent to to OC-SVM for the RBF kernel. OC-SVM are both granted an unfair advantage by selecting its hyper-parameters to maximize the AUROC on a subset (10%) of the test set to establish a strong baseline.

**Isolation Forest (IF)**    Proposed by Liu et al. (2008), IF is a tree-based method that explicitly isolates anomalies instead of constructing a profile of normal instances and then identify instances that do not conform to the normal profile as anomalies. As recommended in the original paper, the number of trees is set to $t = 100$, and the sub-sampling size to $\psi = 256$.

**Kernel Density Estimator (KDE)**    The bandwidth $h$ of the Gaussian Kernel is selected via 5-fold cross-validation using the log-likelihood following Ruff et al. (2018).

| Data | $\gamma_l$ | OC-SVM Raw | OC-SVM Hybrid | IF Raw | IF Hybrid | KDE Raw | KDE Hybrid | CAE | Deep SVDD | Inclusive NRF | SSAD Raw | SSAD Hybrid | SS-DGM | Deep SAD | Supervised Classifier | MML VAE | DP VAE |
|---|---|---|---|---|---|---|---|---|---|---|---|---|---|---|---|---|---|
| MNIST | .00 | 96.0±2.9 | 96.3±2.5 | 85.4±8.7 | 90.5±5.3 | 95.0±3.3 | 87.8±5.6 | 92.9±5.7 | 92.8±4.9 | 95.26±3.0 | 96.0±2.9 | 96.3±2.5 | | 92.8±4.9 | | 94.2±3.0 | 94.2±3.0 |
| | .01 | | | | | | | | | | 96.6±2.4 | 96.8±2.3 | 89.9±9.2 | 96.4±2.7 | 92.8±5.5 | 97.3±2.1 | 97.0±2.3 |
| | .05 | | | | | | | | | | 93.3±3.6 | 97.4±2.0 | 92.2±5.6 | 96.7±2.4 | 94.5±4.6 | 97.8±1.6 | 97.5±2.0 |
| | .10 | | | | | | | | | | 90.7±4.4 | 97.6±1.7 | 91.6±5.5 | 96.9±2.3 | 95.0±4.7 | 97.8±1.6 | 97.6±2.1 |
| | .20 | | | | | | | | | | 87.2±5.6 | 97.8±1.5 | 91.2±5.6 | 96.9±2.4 | 95.6±4.4 | 97.9±1.6 | 97.9±1.8 |
| F-MNIST | .00 | 92.8±4.7 | 91.2±4.7 | 91.6±5.5 | 82.5±8.1 | 92.0±4.9 | 69.7±14.4 | 90.2±5.8 | 89.2±6.2 | | 92.8±4.7 | 91.2±4.7 | | 89.2±6.2 | | 90.8±4.6 | 90.8±4.6 |
| | .01 | | | | | | | | | | 92.1±5.0 | 89.4±6.0 | 65.1±16.3 | 90.0±6.4 | 74.4±13.6 | 91.2±6.6 | 90.9±6.7 |
| | .05 | | | | | | | | | | 88.3±6.2 | 90.5±5.9 | 71.4±12.7 | 90.5±6.5 | 76.8±13.2 | 91.6±6.3 | 92.2±4.6 |
| | .10 | | | | | | | | | | 85.5±7.1 | 91.0±5.6 | 72.9±12.2 | 91.3±6.0 | 79.0±12.3 | 91.7±6.4 | 91.7±6.0 |
| | .20 | | | | | | | | | | 82.0±8.0 | 89.7±6.6 | 74.7±13.5 | 91.0±5.5 | 81.4±12.0 | 91.9±6.0 | 92.1±5.7 |
| CIFAR-10 | .00 | 62.0±10.6 | 63.8±9.0 | 60.0±10.0 | 59.9±6.7 | 59.9±11.7 | 56.1±10.2 | 56.2±13.2 | 60.9±9.4 | 70.0±4.9 | 62.0±10.6 | 63.8±9.0 | | 60.9±9.4 [60.7±9.4] | | 52.7±10.7 | 52.7±10.7 |
| | .01 | | | | | | | | | | 73.0±8.0 | 70.5±8.3 | 49.7±1.7 | 72.6±7.4 [71.7±8.4] | 55.6±5.0 | 73.7±7.3 | 74.5±8.4 |
| | .05 | | | | | | | | | | 71.5±8.1 | 73.3±8.4 | 50.8±4.7 | 77.9±7.2 [77.7±7.8] | 63.5±8.0 | 79.3±7.2 | 79.1±8.0 |
| | .10 | | | | | | | | | | 70.1±8.1 | 74.0±8.1 | 52.0±5.5 | 79.8±7.1 [79.6±7.4] | 67.7±9.6 | 80.8±7.7 | 81.1±8.1 |
| | .20 | | | | | | | | | | 67.4±8.8 | 74.5±8.0 | 53.2±6.7 | 81.9±7.0 [81.2±7.5] | 80.5±5.9 | 82.6±7.2 | 82.8±7.3 |

Table 5: Complete results of the experiment where we increase the ratio of labeled anomalies $\gamma_l$ in the training set. We report the avg. AUROC with st.dev. computed over 90 experiments at various ratios $\gamma_l$. In parenthesis, the results of an ensemble of Deep SAD models ($K = 5$).

| Data | OC-SVM Raw | OC-SVM Hybrid | CAE | Deep SVDD | SSAD Raw | SSAD Hybrid | SS-DGM | Deep SAD | Supervised Classifier | VAE | MML VAE | DP VAE | MML-DP VAE |
|---|---|---|---|---|---|---|---|---|---|---|---|---|---|
| arrhythmia | 84.5±3.9 | 76.7±6.2 | 74.0±7.5 | 74.6±9.0 | 86.7±4.0 | 78.3±5.1 | 50.3±9.8 | 75.9±8.7 | 39.2±9.5 | 85.6 ± 2.4 | 85.7 ± 2.3 | 86.7 ± 1.7 | 87.3 ± 1.7 |
| cardio | 98.5±0.3 | 82.8±9.3 | 94.3±2.0 | 84.8±3.6 | 98.8±0.3 | 86.3±5.8 | 66.2±14.3 | 95.0±1.6 | 83.2±9.6 | 95.5 ± 0.8 | 99.3 ± 0.3 | 99.1 ± 0.4 | 99.2 ± 0.4 |
| satellite | 95.1±0.2 | 68.6±4.8 | 80.0±1.7 | 79.8±4.1 | 96.2±0.3 | 86.9±2.8 | 57.4±6.4 | 91.5±1.1 | 87.2±2.1 | 77.7 ± 1.0 | 92.0 ± 1.5 | 89.2 ± 1.6 | 91.7 ± 1.5 |
| satimage-2 | 99.4±0.8 | 96.7±2.1 | 99.9±0.0 | 98.3±1.4 | 99.9±0.1 | 96.8±2.1 | 99.2±0.6 | 99.9±0.1 | 99.9±0.1 | 99.6 ± 0.6 | 99.7 ± 0.3 | 99.9 ± 0.1 | 99.8 ± 0.1 |
| shuttle | 99.4±0.9 | 94.1±9.5 | 98.2±1.2 | 86.3±7.5 | 99.6±0.5 | 97.7±1.0 | 97.9±0.3 | 98.4±0.9 | 95.1±8.0 | 98.1 ± 0.6 | 99.8 ± 0.1 | 99.9 ± 0.04 | 99.9± 0.09 |
| thyroid | 98.3±0.9 | 91.2±4.0 | 75.2±10.2 | 72.0±9.7 | 97.9±1.9 | 95.3±3.1 | 72.7±12.0 | 98.6±0.9 | 97.8±2.6 | 86.9 ± 3.7 | 99.9 ± 0.04 | 99.9 ± 0.04 | 99.9± 0.03 |

Table 6: Complete results on classic AD benchmark datasets in the setting with a ratio of labeled anomalies of $\gamma_l = 0.01$ in the training set. We report the avg. AUROC with st.dev. computed over 10 seeds.

**Semi-Supervised Anomaly Detection (SSAD)** A kernel method suggested by Görnitz et al. (2013) which is a generalization to SVDD to both labeled and unlabled examples. Also granted the same unfair adavantage as OC-SVM/SVDD.

**Convolutional Autoencoder (CAE)** Autoencoders with convolution and deconvolution layers in the encoder and decoder, respectively. We use the same architectures described in A.3.1 for our VAE.

**Hybrid Methods** In all of the the hybrid methods mentioned in the results, the inputs are representations from a converged autoencoder, instead of raw inputs.

**Unsupervised Deep SVDD** Two end-to-end variants of OC-SVM methods called Soft-Boundary Deep SVDD and One-Class Deep SVDD proposed by Ruff et al. (2018). They use an objective similar to that of the classic SVDD to optimize the weights of a deep architecture.

**Deep Semi-supervised Anomaly Detection (Deep SAD)** Recently proposed by Ruff et al. (2019), Deep SAD is a general method based on deep SVDD, which learns a neural-network mapping of the input that minimizes the volume of data around a predetermined point.

**Semi-Supervised Deep Generative Models (SS-DGM)** Kingma et al. (2014) proposed a deep variational generative approach to semi-supervised learning. In this approach, a classifier is trained on the latent space embeddings of a VAE, which are lower in dimension than the original input. They also propose a probabilistic model that describes the data using the available labels. These two models are fused together to a stacked semi-supervised model.

**Deep Structured Energy-Based Models (DSEBM)** Proposed by Zhai et al. (2016), DSEBM is a deep neural technique, whose output is the energy function (negative log probability) associated with an input sample. The chosen architecture, as decribed by Golan & El-Yaniv (2018) is the same as that of the encoder part in the convolutional autoencoder used by OC-SVM Hybrid.

**Deep Autoencoding Gaussian Mixture Model (DAGMM)** Proposed by Zong et al. (2018), DAGMM is an end-to-end deep neural network that leverages Gaussian Mixture Modeling to perform density estimation and unsupervised anomaly detection in a low-dimensional space learned by deep autoencoder. It simultaneously optimizes the parameters of the autoencoder and the mixture model in an end-to-end fashion, thus leveraging a separate estimation network to facilitate the parameter learning of the mixture model. The architecture, as decribed by Golan & El-Yaniv (2018) is the same as of the autoencoder we used is similar to that of the convolutional autoencoder used in OC-SVM Hybrid.

**Anomaly Detection with a Generative Adversarial Network (ADGAN)** A GAN-based model, proposed by Deecke et al. (2018). Anomaly detection is done with GANs by searching the generator's latent space for good sample representations. In the experiments performed by Golan & El-Yaniv (2018), the generative model of the ADGAN had the same architecture used by the authors of the original paper.

**Deep Anomaly Detection using Geometric Transformations (DADGT)** A stae-of-the-art deep anomaly detection in images method proposed by Golan & El-Yaniv (2018). In this method, features are learned using a self-supervised paradigm – by applying geometric transformations to the image and learning to classify which transformation was applied.

**Inclusive Neural Random Fields (Inclusive-NRF)** A state-of-the-art energy-based model proposed by Song & Ou (2018). The inclusive-NRF learns neural random fields for continuous data by developing inclusive-divergence minimized auxiliary generators and stochastic gradient sampling. As this model directly provides a density estimate, it is an efficient tool for AD, as the estimated density can be used as the decision criterion.

### A.2.1 COMPLETE ABLATIVE ANALYSIS

The complete ablative analysis is reported in table 7.

| Encoder | Decoder | Ensemble | Cardio AUROC | Satellite AUROC | CIFAR-10 AUROC |
|---|---|---|---|---|---|
| Separate | Unfreeze | 5 | 96.6±0.6 | 65.3±1.1 | 49.8±10.7 |
| Separate | Freeze | 5 | 96.6±0.6 | 66.9±1.2 | 50.0±10.9 |
| Same | Unfreeze | 5 | 98.8±0.06 | 88.9±1.7 | 73.9±8.9 |
| Same | Freeze | 5 | **99.1±0.4** | **91.7±1.5** | **74.5±8.4** |
| Separate | Unfreeze | 1 | 82.5±3.3 | 63.1±5.3 | 50.4±10.2 |
| Separate | Freeze | 1 | 88.8±0.9 | 66.2±3.4 | 49.2±10.4 |
| Same | Unfreeze | 1 | 96.8±1.5 | 90.0±2.9 | 72.8±8.6 |
| Same | Freeze | 1 | 97.8±1.1 | 87.6±1.7 | 72.3±9.0 |

Table 7: Complete ablative analysis of the Dual Prior method. AUROC is reported over average of 10 seeds for the *satellite* and *cardio*. For *CIFAR-10*, results are reported for the experiment with 1% of anomalies, averaged over 90 experiments.

### A.3 IMPLEMENTATION DETAILS

We provide essential implementation information and describe how we tuned our models[4].

### A.3.1 NETWORK ARCHITECTURES

Our work is based on deep variational autoencoders which require an encoder network and a decoder network. For the architectures, we follow Ruff et al. (2019) architectures for the autoencoders, with the only differences being the use of bias weights in our architectures and instead of autoencoders we use *variational* autoencoders (i.e. the encoder outputs mean and standard deviation of the latent variables).

For the image datasets, LeNet-type convolutional neural networks (CNNs) are used. Each convolutional layer is followed by batch normalization and leaky ReLU activations ($\alpha = 0.1$) and $2 \times 2$-max pooling. On Fashion-MNIST, we use two convolutional layers, one the first with $16 \times (5 \times 5)$ filters and second with $32 \times (5 \times 5)$ filters. Following are two dense layers of 64 and 32 units respectively. On CIFAR-10, three convolutional layers of sizes $32 \times (5 \times 5)$, $64 \times (5 \times 5)$ and $128 \times (5 \times 5)$ filters are followed by a final dense layer of 128 units (i.e. the latent space is of dimension 128). For MNIST, we use a different architecture where the first layer is comprised of $64 \times (4 \times 4)$-filters and the second $128 \times (4 \times 4)$-filter with ReLU activations and no batch normalization. We use two dense layers, the first with 1024 units and the final has 32 units. For CatsVsDogs we follow Golan & El-Yaniv (2018) architecture for the autoencoder. We employ three convolutional layers, each followed by batch normalization and ReLU activation. The number of filters in each layer are $128 \times (3 \times 3)$, $256 \times (3 \times 3)$ and $512 \times (3 \times 3)$ respectively. The final dense layer has 256 units, which is the latent space dimension.

For the classic AD benchmark datasets, we use regular multi-layer perceptrons (MLPs). On arrhythmia, a 3-layer MLP with 128-64-32 units. On cardio, satellite, satimage-2 and shuttle, we use a 3-layer MLP with 32-16-8 units. On thyroid a 3-layer MLP with 32-16-4 units.

For the motion planning, we follow Ichter et al. (2018) and use two-layer MLP with 512 hidden units and latent space dimension of 256. We use dropout ($p = 0.5$) as regularization and ReLU activation for the hidden layers.

### A.3.2 HYPER-PARAMETERS AND TUNING

Tuning models in a semi-supervised setting is not a trivial task, as usually there is abundance of data from one class and a small pool of data from the other classes. Thus, it is not clear whether one should allocate a validation set out of the training set (and by doing that, reducing the number of available samples for training) or just evaluate the performance on the training set and hope for the best. Ruff et al. (2019) didn't use a validation set, but predetermined the total number of epochs to run and finally evaluated the performance on the test set. We, on the other hand, decided to take a

---

[4]Our code will be published upon acceptance for publication

validation set out of the training set for the image datasets, as we have enough data. The validation set is composed of unseen samples from the normal class and samples from the *current* outlier class unlike the test set, which is composed of samples from all ten classes (9 outlier classes). On the classic AD benchmark datasets, as there is very few outlier data, we evaluate the performance during training on the training set itself without taking a validation set. Finally, we evaluate the performance on the test set. For all datasets, we used a batch size of 128 and an ensemble of 5 VAEs. For the image datasets, we run 200 epochs and for the classical AD benchmarks we run 150 epochs. For the MML method, we found $\gamma = 1$ to perform well on all datasets. Furthermore, similarly to the additional hyper-parameter $\beta_{KL}$ in the ELBO term, we add $\beta_{CUBO}$ to the CUBO term as derived in Equation 4. The motivation for adding this balancing hyper-parameter is that since $\mathcal{L} = \exp\{n \cdot CUBO_n(q_\phi(z|x))\}$ reaches the same optima as $CUBO_n(q_\phi(z|x))$ as explained earlier, so does $\mathcal{L} = \exp\{\beta \cdot n \cdot CUBO_n(q_\phi(z|x))\}$. Finding a good value for $\beta_{CUBO}$ can contribute to the performance, as reflected in the results. Moreover, for the optimization of the CUBO component, we used gradient clipping and learning rate scheduling (every 50 epochs, learning rate is multiplied by 0.1). Tables 8 and 9 summarize all the hyper-parameters chosen for our model per dataset.

| Data | ND Update Interval | Learning Rate | $\beta_{KL}$ | $\mu_o$ |
|------|-------------------|---------------|--------------|---------|
| MNIST | 2 | 0.001 | 0.005 | 10 |
| Fashion-MNIST | 2 | 0.001 | 0.005 | 10 |
| CIFAR-10 | 2 | 0.001 | 0.005 | 10 |
| CatsVsDogs | 1 | 0.001 | 0.005 | 10 |
| arrhythmia | 1 | 0.0005 | 0.5 | 2 |
| cardio | 1 | 0.001 | 0.05 | 5 |
| satellite | 1 | 0.001 | 0.05 | 5 |
| satimage-2 | 1 | 0.0005 | 0.05 | 10 |
| shuttle | 1 | 0.001 | 0.05 | 5 |
| thyroid | 1 | 0.0001 | 0.05 | 10 |

Table 8: Hyper-parameters for the Dual Prior VAE

| Data | ND Update Interval | Learning Rate | $\beta_{KL}$ | $\beta_{CUBO}$ |
|------|-------------------|---------------|--------------|----------------|
| MNIST | 2 | 0.0005 | 0.005 | 0.005 |
| Fashion-MNIST | 2 | 0.001 | 0.005 | 0.005 |
| CIFAR-10 | 2 | 0.001 | 0.005 | 0.005 |
| CatsVsDogs | 1 | 0.0005 | 0.005 | 0.005 |
| arrhythmia | 1 | 0.0005 | 0.5 | 0.5 |
| cardio | 1 | 0.001 | 0.05 | 0.05 |
| satellite | 1 | 0.001 | 0.05 | 0.05 |
| satimage-2 | 1 | 0.001 | 0.05 | 0.05 |
| shuttle | 1 | 0.001 | 0.05 | 0.05 |
| thyroid | 1 | 0.0001 | 0.05 | 0.05 |

Table 9: Hyper-parameters for the Max-Min Likelihood VAE

For training, we follow Kaae Sønderby et al. (2016) recommendations for training VAEs and use a 20-epoch annealing for the KL-divergence component, that is, the KL-divergence coefficient, $\beta_{KL}$, is linearly increased from zero to its final value. Moreover, we allow the VAE to first learn a good representation of the normal data in a warm-up period of 50 epochs, and then we begin applying the novelty detection updates. For optimization, we use Adam (Kingma & Ba, 2014) with a learning rate schedule to stabilize training when outlier samples are fed after the warm-up period.

## A.4 CUBO LOSS DERIVATION

The $\chi$-divergence is defined as follows:

$$D_{\chi^2}(p\|q) = \mathbb{E}_{q(z;\theta)}\Big[\big(\frac{p(z|x)}{q(z;\theta)}\big)^2 - 1\Big]$$

The general upper bound as derived in Dieng et al. (2017):

$$\mathcal{L}_{\chi^2}(\theta) = CUBO_2 = \frac{1}{2}\log \mathbb{E}_{q(z;\theta)}\Big[\big(\frac{p(x,z)}{q(z;\theta)}\big)^2\Big]$$

The optimized CUBO:

$$\mathcal{L} = \exp\{2 \cdot CUBO_2(\theta)\} = \mathbb{E}_{q(z;\theta)}\left[\left(\frac{p(x,z)}{q(z;\theta)}\right)^2\right]$$

In the our VAE framework, we denote $\mathcal{L}_R = -\log p(X|z)$, the reconstruction error between the output of the decoder and the original input. We assume that $q(z|X) \sim \mathcal{N}(\mu_q(X), \Sigma_q(X))$ and that in general, $p_{outlier}(z) \sim \mathcal{N}(\mu_o, I)$, but we note that in our case, we set $\mu_o = 0$, for both CUBO and hybrid methods. As the CUBO loss is derived only for the anomalous data, we omit the labels in the following. $\beta_{CUBO}$ is a balancing hyper-parameter we add, as mentioned in A.3.2.

$$\mathcal{L}_{CUBO} = \mathbb{E}_{q(z;\theta)}\left[\left(\frac{p(x,z)}{q(z;\theta)}\right)^2\right] = \mathbb{E}_q\left[\exp\{2\log p(X|z) + 2\log\left(\frac{p(z)}{q(z|X)}\right)\}\right] =$$

$$\mathbb{E}_q\left[\exp\{-2 \cdot \mathcal{L}_R + 2\beta_{CUBO} \cdot (\log p(z) - \log q(z|X))\}\right] =$$

$$\mathbb{E}_q\left[\exp\{-2\cdot\mathcal{L}_R + 2\beta_{CUBO}\cdot\left(-\frac{1}{2}(z-\mu_o)^T(z-\mu_o) - [-\frac{1}{2}\log|\Sigma_q| - \frac{1}{2}(z-\mu_q)^T\Sigma_q^{-1}(z-\mu_q)]\right)\}\right] =$$

$$\mathbb{E}_q\left[\exp\{-2 \cdot \mathcal{L}_R + \beta_{CUBO} \cdot \left(-[z^Tz - 2z^T\mu_o + \mu_o^T\mu_o] + \log|\Sigma_q| + \right.\right.$$
$$\left.\left. z^T\Sigma_q^{-1}z - 2z^T\Sigma_q^{-1}\mu_q + \mu_q^T\Sigma_q^{-1}\mu_q\right)\}\right] =$$

$$\mathbb{E}_q\left[\exp\{-2 \cdot \mathcal{L}_R + \beta_{CUBO} \cdot \left(\log|\Sigma_q| + \mu_q^T\Sigma_q^{-1}\mu_q - \mu_o^T\mu_o - \right.\right.$$
$$\left.\left. z^Tz + 2z^T\mu_o + z^T\Sigma_q^{-1}z - 2z^T\Sigma_q^{-1}\mu_q\right)\}\right] =$$

$$\exp\{-2 \cdot \mathcal{L}_R + \beta_{CUBO} \cdot \left(\log|\Sigma_q| + \mu_q^T\Sigma_q^{-1}\mu_q - \mu_o^T\mu_o\right)\} \cdot$$
$$\mathbb{E}_q\left[\exp\{\beta_{CUBO} \cdot \left(-z^Tz + 2z^T\mu_o + z^T\Sigma_q^{-1}z - 2z^T\Sigma_q^{-1}\mu_q\right)\}\right] =$$

$$\exp\{-2 \cdot \mathcal{L}_R + \beta_{CUBO} \cdot \left(\log|\Sigma_q| + \mu_q^T\Sigma_q^{-1}\mu_q - \mu_o^T\mu_o\right)\} \cdot$$
$$\exp\{\log\mathbb{E}_q\left[\exp\{\beta_{CUBO} \cdot \left(-z^Tz + 2z^T\mu_o + z^T\Sigma_q^{-1}z - 2z^T\Sigma_q^{-1}\mu_q\right)\}\right]\} =$$

$$\exp\{-2 \cdot \mathcal{L}_R + \beta_{CUBO} \cdot \left(\log|\Sigma_q| + \mu_q^T\Sigma_q^{-1}\mu_q - \mu_o^T\mu_o\right) +$$
$$\log\mathbb{E}_q\left[\exp\{\beta_{CUBO} \cdot \left(-z^Tz + 2z^T\mu_o + z^T\Sigma_q^{-1}z - 2z^T\Sigma_q^{-1}\mu_q\right)\}\right]\} \quad (4)$$

The expectation is estimated with Monte Carlo and for numeric stability we employ the commonly used *log-sum-exp trick*.

## A.5 ELBO WITH GAUSSIAN PRIOR

We provide the derivation for the ELBO objective function where the prior, $p(z)$ is a Gaussian with non-zero mean, that is, $z \sim \mathcal{N}(\mu_o, I)$. Under the i.i.d. assumption in the VAE framework, we assume that $z_i \sim \mathcal{N}(\mu_{o_i}, 1)$ and $z_i|x \sim \mathcal{N}(\mu_{q_i}, \sigma_{ii}^2)$. Thus, it holds that:

$$\mathbb{E}_q[z_i^2] = \sigma_{ii}^2 + \mu_{q_i}^2$$

We now derive the KL-divergence component of the ELBO:

$$D_{\mathrm{KL}}[q(z|X)\|p(z)] = \mathbb{E}_q\left[\frac{\log q(z|X)}{\log p(z)}\right] = \mathbb{E}_q[\log q(z|X)] - \mathbb{E}_q[\log p(z)]$$

$$\mathbb{E}_q[\log q(z|X)] = -\frac{1}{2}\log|\Sigma_q| - \frac{1}{2}\mathbb{E}_q[(z-\mu_q)^T\Sigma_q^{-1}(z-\mu_q)] =$$

$$-\frac{1}{2}\log|\Sigma_q| + \frac{1}{2}\mu_q^T\Sigma_q^{-1}\mu_q - \frac{1}{2}\mathbb{E}_q[z^T\Sigma_q^{-1}z] =$$

$$-\frac{1}{2}\sum_{i=1}^{n}[\log\sigma_{ii}^2 - \frac{\mu_{q_i}^2}{\sigma_{ii}^2} + 1 + \frac{\mu_{q_i}^2}{\sigma_{ii}^2}] = -\frac{1}{2}\sum_{i=1}^{n}[\log\sigma_{ii}^2 + 1]$$

$$\mathbb{E}_q[\log p(z)] = -\frac{1}{2}\mathbb{E}_q[(z - \mu_o)^T(z - \mu_o)] =$$

$$\frac{1}{2}\sum_{i=1}^{n}[\sigma_{ii}^2 + \mu_{q_i}^2 - 2\mu_{q_i}\mu_{o_i} + \mu_{o_i}^2]$$

Finally:

$$D_{\mathrm{KL}}[q(z|X)\|p(z)] = -\frac{1}{2}\sum_{i=1}^{n}[1 + \log\sigma_{ii}^2 - \sigma_{ii}^2 - \mu_{q_i}^2 + 2\mu_{q_i}\mu_{o_i} - \mu_{o_i}^2]$$

