# OpenReview forum: "Deep Variational Semi-Supervised Novelty Detection"
_ICLR.cc/2020/Conference — Reject_

### Official Review · AnonReviewer1 · 2019-10-20
**Official Blind Review #1**

**Rating:** 6

**Review:**

The papers proposes to use VAE-like approaches for semi-supervised novelty detection. Two methods are describes:
(1) the MML-VAE fits a standard VAE to the normal samples and add a repulsive term for the outliers -- this term encourages the encoder to map the outliers far from the latent-space prior distribution.
(2) the DP-VAE fits a VAE with a mixture of Gaussian prior on the latent space -- one mixture component for normal samples, and one mixture component for outlier samples.

The described methods are simple, natural, and appear to work relatively well -- for this simple reason, I think that the text could be accepted.

There are several things that are still not entirely clear.
(1) without the reconstruction term, the methods are small variations of supervised methods. Consequently, I feel that the authors should try to explain much more carefully why the introduction of a reconstruction term (which could be thought as an auxiliary task) helps.
(2) given point (1), one could think of many auxiliary task (eg. usual colorisation, or rotation prediction, etc..) Would it lead to worse results?
(3) proportion > 10% of anomaly is relatively standard for supervised-methods + few standard tricks to work very well. Although I understand that only one small subset of anomalies is presented during training, I think that it would still be worth describing in more details the efforts that have been spent to try to make standard supervised methods work.


**Experience Assessment:**

I have read many papers in this area.

**Review Assessment: Checking Correctness Of Derivations And Theory:**

I carefully checked the derivations and theory.

**Review Assessment: Checking Correctness Of Experiments:**

I assessed the sensibility of the experiments.

**Review Assessment: Thoroughness In Paper Reading:**

I read the paper thoroughly.

---

> ### Author Response · Authors · 2019-11-11
> **Addressing the reviewer concerns**
>
> Dear reviewer, thank you for the encouraging review. We hope that the following will alleviate your concerns:
>
> 1. The reconstruction term in the VAE is a direct consequence of optimizing the variational lower bound of the data log-likelihood. According to this formulation, our model is trained to output the (approximate) likelihood of data samples, making for a principled approach to novelty detection: a test sample is detected as normal if its likelihood under our model is high.
> This is starkly different from discriminative supervised methods, which do not model the data distribution. The generative unsupervised approach is fundamentally sound *even without any labeled outlier data*, and our work shows that just a few labeled samples can significantly improve results.
>
> 2. We share similar views on improving anomaly detection with other auxiliary tasks. For example, [1], the SOTA in AD for images uses self-supervision with geometric transformations. It would be interesting to combine such approaches with our work, and we are actively working on this direction.
>
> 3. We emphasize that our method works well even with much less than 10% of anomalies (e.g., 1% can lead to dramatic improvement in our results).
> While there are tricks for mitigating class imbalance for supervised methods, we emphasize that our experimental setting, which builds on [2], is very different: it measures novelty detection of *unknown classes*. That is, during training, the classifier sees a large proportion of data from the normal class (e.g., the 0 digit in MNIST), and a small proportion from *only one* of the other anomaly classes (e.g., the digit 3). At test time, the anomalies are presented *from all of the classes* (digits 1-9). Thus, any class imbalance trick applied to supervised learning will not help, and we are not aware of standard tricks to make a discriminatively trained network correctly generalize to classes it has never seen before. This is validated in our results: the supervised classifiers failed to generalize to the unseen anomaly classes.
>
> [1] Golan, Izhak, and Ran El-Yaniv. "Deep anomaly detection using geometric transformations." Advances in Neural Information Processing Systems. 2018.
> [2] Ruff, Lukas, et al. "Deep Semi-Supervised Anomaly Detection." arXiv preprint arXiv:1906.02694 (2019).

---

### Official Review · AnonReviewer3 · 2019-10-21
**Official Blind Review #3**

**Rating:** 3

**Review:**

This paper presents two novel VAE-based methods for the (more general) semi-supervised anomaly detection (SSAD) setting where one has also access to some labeled anomalous samples in addition to mostly normal data. The first method, Max-Min Likelihood VAE (MML-VAE), extends the standard VAE objective that maximizes the log-likelihood for normal data by an additional term that in contrast minimizes the log-likelihood for labeled anomalies. To optimize the MML objective, the paper proposes to minimize the sum of the standard (negative) ELBO for normal samples and the so-called CUBO, which is a variational upper bound on the data log-likelihood, for anomalous samples. The second method, Dual Prior VAE (DP-VAE), modifies the standard VAE by introducing a second separate prior for the anomalous data, which is also Gaussian but has different mean. The DP-VAE objective then is defined as the sum of the two respective ELBOs which is optimized over shared encoder and decoder networks (with the adjustment that the outlier ELBO only updates the encoder). The anomaly score for both models then is defined as the (negative) ELBO of a test sample. Finally, the paper presents quite extensive experimental results on the benchmarks from Ruff et al. [2], CatsVsDogs, and an application of robotic motion planning which indicate a slight advantage of the proposed methods.

I am quite familiar with the recent Deep SAD paper [2] this work builds upon and very much agree that the (more general) SSAD setting is an important problem with high practical relevance for which there exists little prior work. Overall this paper is well structured/written and well placed in the literature, but I think it is not yet ready for acceptance due to the following key reasons:
(i) I think DP-VAE, the currently better performing method, is ill-posed for SSAD since it makes the assumption that anomalies are generated from one common latent prior and thus must be similar;
(ii) I think the worse performance of MML-VAE, which I find theoretically sound for SSAD, is mainly due to optimization issues that should be investigated;
(iii) The experiments do not show for the bulk of experiments how much of the improvement is due to meta-algorithms (ensemble and hyperparameter selection on a validation set with some labels).

(i) DP-VAE models anomalies to be generated from one common latent distribution (modeled as Gaussian here) which imposes the assumption that anomalies are similar, the so-called cluster assumption [2]. This assumption, however, generally does not hold for anomalies which are defined to be just different from the normal class but anomalies do not have to be similar to each other. Methodologically, DP-VAE is rather a semi-supervised classification method (essentially a VAE with Gaussian mixture prior having two components) which the paper itself points out is ill-posed for SSAD: “... the labeled information on anomalous samples is too limited to represent the variation of anomalies ... .” I suspect the slight advantage of DP-VAE might be mainly due to using meta-algorithms (ensemble, hyperparameter selection) and due to the rather structured/clustered nature of anomalies in the MNIST, F-MNIST, and CIFAR-10 benchmarks.

(ii) I find MML-VAE, unfortunately the worse performing method, to be a conceptually sound approach to SSAD following the intuitive idea that normal samples should concentrate under the normal prior whereas the latent embeddings of anomalies should have low likelihood under this prior. This approach correctly does not make any assumption on the latent structure of anomalies as DP-VAE does. I believe MML-VAE in its current formulation leads to worse results mainly to optimization issues that I suspect can be resolved and should be further investigated. I guess the major issue of the MML-VAE loss is that the log-likelihood for outlier samples has steep curvature and is unbounded from below. Deep networks might easily exploit this without learning meaningful representations as the paper also hints towards. This also results in unstable optimization. I think removing the reconstruction term for outliers, as the paper suggests, also helps for this particular reason but this is rather heuristic. These optimization flaws should be investigated and the loss adjusted if needed. Maybe simple thresholding (adding an epsilon to lower bound the loss), gradient clipping, or robust reformulations of the loss could improve optimization already?

(iii) To infer the statistical significance of the results and to assess the effect of meta-algorithms (ensemble, hyperparameter tuning) an ablation study as in Table 4 (at least on the effect of ensembling) should be included also for the major, more complex datasets. Which score is used for hyperparameter selection (ELBO, log-likelihood, AUC)? How would the competitors perform under similar tuning?


####################
*Additional Feedback*

*Positive Highlights*
1. Both proposed methods can be used with general data types and VAE network architectures (the existing Deep SAD state-of-the-art method employs restricted architectures).
2. The paper is well placed in the literature and all major and very recent relevant work that I am aware of are included.
3. This is an interesting use of the CUBO bound which I did not know before reading this work. This might be interesting for the general variational inference community to derive novel optimization schemes.
4. I found the robotic motion planning application quite cool. This also suggests that negative sampling is useful beyond the AD task.
5. I appreciate that the authors included the CatsVsDogs experiment although DADGT performs better as it demonstrates the potential of SSAD. I very much agree that employing similar self-supervised learning ideas and augmentation is a promising direction for future research.

*Ideas for Improvement*
6. Extend the semi-supervised setting to unlabeled (mostly normal), labeled normal, and labeled anomalous training data. The text currently formulates a setting with only labeled normal and labeled anomalous samples. A simple general formulation could just assign different weights to the unlabeled and labeled normal data terms.
7. There might be an interesting connection between MML-VAE and Deep SAD in the sense that MML-VAE is a probabilistic version of the latter. The $\chi_n$ distance of the CUBO loss has terms similar to the inverse squared norm penalty of Deep SAD.
8. Report the range from which hyperparameters are selected.
9. Add the recently introduced MVTec AD benchmark dataset to your experimental evaluation [1].
10. Run experiments on the full test suite of Ruff et al. [2]. At the moment only one of three scenarios are evaluated.

*Minor comments*
11. Inconsistent notation for the expected value ($\mathbb{E}$ vs $\mathbf{E}$)
12. In Section 3, the parameterization of the variational approximate $q(z | x)$ is inconsistently denoted by $\phi$ and $\theta$ (which beforehand parameterizes the decoder).
13. In Section 3.2, the current formulation first says that MC produces a biased, then an unbiased estimate of the gradients.
14. First sentence in Section 4: I would not use “classify” but rather “detect” etc. for anomaly/novelty detection since the task differs from classification.
15. In Section 4.2, there should be a minus in front of the KL-divergence terms of the $ELBO_{normal}$ and $ELBO_{outlier}$ equations.
16. In the fully unsupervised setting on CIFAR-10 (Table 5), why is the VAE performance essentially at random (~50) in comparison to CAE and Deep SVDD although they use the same network architecture?
17. Is the CUBO indeed a strictly valid bound if one considers the non-normal data-generating distribution?
18. Are there any results on the tightness of the CUBO?


####################
*References*
[1] P. Bergmann, M. Fauser, D. Sattlegger, and C. Steger. Mvtec ad–a comprehensive real-world dataset for unsupervised anomaly detection. In Proceedings of the IEEE Conference on Computer Vision and Pattern Recognition, pages 9592–9600, 2019.
[2] L. Ruff, R. A. Vandermeulen, N. Görnitz, A. Binder, E. Müller, K.-R. Müller, and M. Kloft. Deep semi-supervised anomaly detection. arXiv preprint arXiv:1906.02694, 2019.

**Experience Assessment:**

I have published in this field for several years.

**Review Assessment: Checking Correctness Of Derivations And Theory:**

I assessed the sensibility of the derivations and theory.

**Review Assessment: Checking Correctness Of Experiments:**

I carefully checked the experiments.

**Review Assessment: Thoroughness In Paper Reading:**

I read the paper thoroughly.

---

> ### Author Response · Authors · 2019-11-11
> **Addressing the reviewer concerns**
>
> Dear reviewer, thank you for your extensive review. We greatly appreciate the effort to improve our paper.
>
> 1. While DP-VAE builds on a latent distribution for anomalies, this does not imply that anomalies are similar!
> For example, VAEs are known to generate very expressive distributions from just a single Gaussian prior.
> Furthermore, note that even a standard VAE, which approximates the data distribution, is already a capable novelty detector just by training on normal data samples.
> The DP-VAE acts to *refine* the anomaly detection, by pushing anomalies outside the normal-data prior, thereby increasing their KL.
> The fact that in DP-VAE the KL increase has a direction (towards another Gaussian), while in MML-VAE it is not directed towards a specific point, does not mean that the method requires outliers to be clustered. This is because, as described in Section 4.3, *we only use the normal data prior for novelty detection*!
> This is also clearly demonstrated in our experiments. We train on data from only one class of anomalies, and test on different classes. If training on the class of, say, dogs, with anomalies from class, say, airplane, improves anomaly detection of images of horses, this clearly indicates that our method *does not overfit to the training anomaly class*.
>
> 2. You are correct. Due to limited computational resources, we were not able to optimize MML-VAE in time for the deadline. During the review period, we were able to tune the model better, and the results are reported in the revised paper. MML-VAE is on par with DP-VAE. We have independently experimented with all of the methods you suggested, including gradient clipping, thresholding and learning rate scheduling. Learning rate scheduling combined with gradient clipping led to the improved results.
>
> 3. We have extended our ablation study to CIFAR-10. Hyperparmeter tuning for the image datasets was done by measuring the AUROC on a validation set taken from training data, i.e., the normal class and the currently-trained-on outlier class (only one class).
> Ensemble ablation study: per our answer to reviewer #2, we agree, this is a valid concern, and we have expanded our experiments to address it. In our study, ensembles can improve our method’s result by approximately 2-4% on average, which makes sense as VAE training is a stochastic process. However, we found that ensembles *do not improve the results for Deep-SAD, the previous state-of-the-art*. This can be explained as follows. In Deep-SAD, confidence is measured according to distance to an arbitrary point C in the vector space. Thus, the scores from different networks in the ensemble are not necessarily calibrated (they have different C points). In our VAE approach, on the other hand, the score of each network is derived from  the same principled variational bound, and therefore the scores are calibrated, giving rise to the benefit of the ensemble.
> To summarize, the use of ensembles is a specific feature of our approach, and using it does not improve the previous state-of-the-art Deep-SAD method, further supporting our claims in the paper.
>
> 4. Thank you for the ideas for improvement. We agree that combining Deep-SAD with the CUBO may prove to be an exciting direction. We are happy to include additional anomaly detection experiments as the reviewer suggested in the final version. The hyperparameters are dependent on the type of data, and the range is similar within the types. We have detailed the hyper-parameters for each dataset in the appendix.
>
> 5. Minor comments section: thank you for noticing the small errors in the text. As for the rest:
> a. Autoencoders and Variational Autoencoders are very different in theory and in practice, and are known to generate different behaviour, even for similar architectures.
>
> b. The CUBO was derived and analyzed in [1]. It is not limited to normal distributions.
>
> Once again, we are grateful for the efforts you made in your review and hope that we have addressed your concerns.
>
> [1] Dieng, Adji Bousso, et al. "Variational Inference via $\chi $ Upper Bound Minimization." Advances in Neural Information Processing Systems. 2017.

---

### Official Review · AnonReviewer2 · 2019-10-27
**Official Blind Review #2**

**Rating:** 6

**Review:**

This paper proposes two variational methods for training VAEs for SSAD (Semi-supervised Anomaly Detection). Experiments on benchmarking datasets show improvements over state-of-the-art SSAD methods.

In generally, the paper is well written. But I have some concerns.

1. Some of the results have not yet been obtained.

2. Missing some relevant references.
In addition to VAEs, there is another class of deep generative models - random fields (a.k.a. energy-based models, EBMs), which have been applied to anomaly detection (AD) recently. Particularly, the unsupervised AD results on MNIST and CIFAR-10 from [2] are much better than the proposed methods (MML-VAE, DP-VAE).
Though semi-supervised AD is interesting, good performances on unsupervised AD can be a baseline indicator of the effectiveness of the AD models. The authors should add comments and comparisons.

[1] S. Zhai, Y. Cheng, W. Lu, and Z. Zhang, “Deep structured energy based models for anomaly detection,” ICML, 2016.
[2] Y. Song, Z. Ou. "Learning Neural Random Fields with Inclusive Auxiliary Generators," arxiv 1806.00271, 2018.

3. “For all of the experiments, our methods use an ensemble of size K = 5.”
Are other methods also tested by using an ensemble?

--------update after reading the response-----------
The updated paper has been improved to address my concerns.

I partly agree with the authors that their results demonstrate the importance of the semi-supervised AD setting (a 1% fraction of labelled anomalies can improve over the state-of-the-art AD scores of deep energy based models). However, I think, the proposed methods in this paper will not be as competitive as semi-supervised deep energy based models.

**Experience Assessment:**

I have published one or two papers in this area.

**Review Assessment: Checking Correctness Of Derivations And Theory:**

I assessed the sensibility of the derivations and theory.

**Review Assessment: Checking Correctness Of Experiments:**

I assessed the sensibility of the experiments.

**Review Assessment: Thoroughness In Paper Reading:**

I read the paper at least twice and used my best judgement in assessing the paper.

---

> ### Author Response · Authors · 2019-11-11
> **Addressing the reviewer concerns**
>
> Dear reviewer, thank you for taking the time to review our paper. Your comments are much appreciated! Regarding your concerns:
>
> 1. All the results are now reported in the revised paper. The missing experiments in the submission were unfortunately due to limited resources - some of the MML-VAE experiments did not finish before the ICLR deadline.
>
> 2. DS-EBM results are reported for the cats-vs-dogs experiment, and described in Appendix A.2. Thank you for pointing us to Song et al.’s recent work. These results are impressive, and we have duly added them to the text.
> That said, we emphasize that *our work focuses on semi-supervised anomaly detection, which has not been handled by said EBM approaches*. Specifically, while for unsupervised AD, Inclusive-NRF’s results of 70.2% on CIFAR are much better than the 52.7% score of the vanilla VAE, even just 1% of labelled data is enough to get 74.5% with our DP-VAE.
> Thus, for the SSAD setting, *our results are state-of-the-art*, and further demonstrate the importance of the SSAD setting.
> We also mention that the Inclusive-NRF work used much more expressive ResNet architectures, while we opted for much simpler architectures, for a fair comparison with the Deep-SAD work (otherwise one could have argued that our improvement is only due to better architecture search).
> We agree that studying SSAD with EBMs is a very promising direction. In any such study, though, our current results should be a baseline for comparison.
> Finally, the promising results of EBMs *should not waive off the study of VAE-based methods*. VAEs are widely used in many applications, and our experiments on motion planning demonstrate the benefit of extending VAEs to SSAD. Since our work is also SOTA in SSAD, we kindly ask the reviewer to reconsider the evaluation of our work’s motivation.
>
> 3. We agree - as reported, our use of ensembles raises a valid concern, and we have expanded our experiments to address it.
> In our study, ensembles can improve our method’s result by approximately 2-4% on average, which makes sense as VAE training is a stochastic process. However, we found that ensembles *do not improve the results for Deep-SAD, the previous state-of-the-art*. This can be explained as follows. In Deep-SAD, confidence is measured according to distance to an arbitrary point C in the vector space. Thus, the scores from different networks in the ensemble are not necessarily calibrated (they have different C points). In our VAE approach, on the other hand, the score of each network is derived from  the same principled variational bound, and therefore the scores are calibrated, giving rise to the benefit of the ensemble.
> To summarize, the use of ensembles is a specific feature of our approach, and using it does not improve the previous state-of-the-art Deep-SAD method, further supporting our claims in the paper.
>
> We thank you again for your comments and hope you will reconsider your evaluation of our paper.

---

### Author Response · Authors · 2019-11-11
**General Statement**

We thank the reviewers for their thoughtful feedback.

We want to emphasize the motivation for our SSAD study: as our experiments demonstrate, even a small fraction of labeled data can significantly improve AD performance. Furthermore, our methods allow to improve general VAEs with negative samples, which, as we show, can have interesting applications beyond AD.

Two reviewers questioned how much the ensembles play a role in our results. We have added experiments to address that, and detailed our findings in our answers below. In short, ensembles are important for our method, but do not help the competing Deep SAD method. Thus, our proposed variational method (+ensembles) is, to our knowledge, the best performing SSAD approach.

We hope that our answers below address the other concerns, and we kindly ask the reviewers to reconsider their scores, given that our method is novel, general, well-motivated, and exhibits state-of-the-art performance.

---

### Decision · Program_Chairs · 2019-12-19

**Decision:**

Reject

**Comment:**

This paper presents two novel VAE-based methods for semi-supervised anomaly detection (SSAD) where one has also access to a small set of labeled anomalous samples. The reviewers had several concerns about the paper, in particular completely addressing reviewer #3's comments would strengthen the paper.